# TTVOS: Lightweight Video Object Segmentation with Adaptive Template Attention Module and Temporal Consistency Loss

## ABSTRACT

*Semi-supervised video object segmentation* (semi-VOS) is widely used in many applications. This task is tracking class-agnostic objects by a given segmentation mask. For doing this, various approaches have been developed based on online-learning, memory networks, and optical flow. These methods show high accuracy but are hard to be utilized in real-world applications due to slow inference time and tremendous complexity. To resolve this problem, template matching methods are devised for fast processing speed, sacrificing lots of performance. We introduce a novel semi-VOS model based on a temple matching method and a novel temporal consistency loss to reduce the performance gap from heavy models while expediting inference time a lot. Our temple matching method consists of short-term and long-term matching. The short-term matching enhances target object localization, while long-term matching improves fine details and handles object shape-changing through the newly proposed adaptive template attention module. However, the long-term matching causes error-propagation due to the inflow of the past estimated results when updating the template. To mitigate this problem, we also propose a temporal consistency loss for better temporal coherence between neighboring frames by adopting the concept of a transition matrix. Our model obtains 79.5% *J&F* score at the speed of 73.8 FPS on the DAVIS16 benchmark.

## CCS CONCEPTS

• **Computing methodologies → Video segmentation**.

## KEYWORDS

Semi-supervised video segmentation, video tracking, lightweight segmentation

## 1 INTRODUCTION

Video object segmentation (VOS) is essential in many applications such as autonomous driving, video editing, and surveillance system. In this paper, we focus on a *semi-supervised video object segmentation* (semi-VOS) task, which is to track a target in a pixel-wise resolution from a given annotated mask for the first frame.

For accurate tracking, many models have been developed, but it is hard to use the models in real-world environment due to tremendous computation. For example, one of popular method, online-learning, fine-tunes model parameters using the first frame image and the corresponding ground truth mask [2, 20, 23, 26]. This strategy makes the model more specialize in each video input, but, it requires additional time and memory for fine-tuning. Memory network method achieves high accuracy than any other approaches. They stacks multiple target memories and match the current frame with the entries. Therefore, the inference time and the required memories increase in proportion to the number of frames. To solve these problems, GC

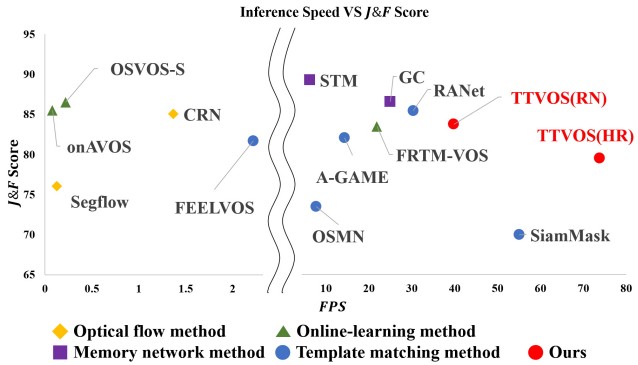

**Figure 1: The speed (FPS) vs accuracy (*J&F* score) on the DAVIS2016 validation set. Our proposed TTVOS achieves high accuracy with small complexity. HR/RN respectively denotes HRNet/ResNet50 for the backbone network.**

[16] conducted weighted-average to the multiple memories at each time frame for generating one global context memory. However, it needs an additional feature extraction step for updating the memory from the current estimated mask and the image. Also, we believe that it is not enough to directly comprehend spatial information since the size of global context memory much smaller than original spatial resolution size.

For increasing consistency of masks across frames, optical flow is one of the popular methods in low-level vision which has been applied in diverse video applications. In a video segmentation task, it propagates a given mask or features by computing pixel-wise trajectories or movements of objects [4, 9, 17, 38]. However, it is too demanding to compute exact flow vectors which contain excessive information for the segmentation task. For example, if we know the binary information of whether a pixel is changed into the foreground or background, we do not need an exact flow vector of each pixel.

The aforementioned methods have increased accuracy a lot, but they require heavy inference time and memory. The template matching approach resolves this problem by designing a target template from a given image and annotation. However, the accuracy is lower compared to other models because the matching method is too simple, and the template is hard to handle object shape variation

In this paper, we propose an adaptive template matching method and a novel temporal consistency loss for semi-VOS. Our contributions can be summarized as follows: 1) We propose a new lightweight VOS model based on template matching method by combining short-term and long-term matching to achieve fast inference time and to reduce the accuracy gap from heavy and complex models. More specifically, in short-term matching, we compare the current frame's feature with the information in the previous frame for localization.

In long-term matching, we devise an adaptive template for generating an accurate mask. 2) We introduce a novel adaptive template motivated from GC for managing shape variation of target objects. Our adaptive template is updated from the current estimated mask without re-extracting features and occupying additional memory. 3) To train the model, we propose a new temporal consistency loss for mitigating the error propagation problem, one of the main reasons for performance degradation, caused by inflow of the past estimated results. To the best of our knowledge, this work is the first to apply the concept of consistency loss for the semi-VOS task without optical flow. Our model generates a transition matrix to encourage the correction of the incorrectly estimated pixels from the previous frame and preventing their propagation to future frames. Our model achieves 79.5% *J&F* score at the speed of 73.8 FPS on the DAVIS16 benchmark (See Fig. 1). We also verified the efficacy of the temporal consistency loss by applying it to other models and showing increased performance.

## 2 RELATED WORK

**Optical flow:** Optical flow which estimates flow vectors of moving objects is widely used in many video applications [7, 12, 28, 31]. In the semi-VOS task, it aligns the given mask or features with the estimated flow vector. Segflow [4] designed two branches, each for image segmentation and optical flow. The outputs of both branches are combined together to estimate the target masks. Similarly, FAVOS [17] and CRN [9] refined a rough segmentation mask by optical flow.

**Online-learning:** The online-learning method is training the model with new data in each inference iteration [14, 27, 45]. In the semi-VOS task, model parameters are fine-tuned in the inference stage with a given input image and a corresponding mask. Therefore, the model is specialized for the given condition of the clip [2, 20, 23]. However, fine-tuning causes additional latency in inference time. [26] resolved this issue by dividing the model into two sub-networks. One is a lightweight network that is fine-tuned in the inference stage for making a coarse score map. The other is a heavy segmentation network without the need for fine-tuning. This network enables fast optimization and relieves the burden of online-learning.

**Memory network:** The memory network constructs external memory representing various properties of the target. It was devised for handling long-term sequential tasks in the natural language processing (NLP) domain, such as the QA task [13, 30, 42]. STM [22] adopted this idea for the semi-VOS task by a new definition of key and value. The *key* encodes visual semantic clue for matching and the *value* stores detailed information for making the mask. However, it requires lots of resources because the amount of memory is increased over time. Furthermore, the size of memory is the square of the resolution of an input feature map. To lower this huge complexity, GC [16] does not stack memory at each time frame, but accumulate them into one, which is also of a smaller size than a unit memory of STM. They does not make a $(hw \times hw)$ memory like [39, 47] but a $(c_{key} \times c_{val})$ memory[1] as similar channel attention module.

**Template matching:** Template matching is one of the traditional method in the tracking task. It generates a template and calculates

---

[1] $h$ and $w$ are the height and the width of an input feature map for constructing memory, and $c_{key}$ and $c_{val}$ are the number channels for the key and value feature maps.

similarity with input as a matching operation. Most works match a feature map from a given image and a template following the siamese network [1], but A-GAME [11] designed a target distribution by a mixture of Gaussian in an embedding space. It predicted posterior class probabilities for matching. RANet [40] applied a racking system to the matching process between multiple templates and input for extracting reliable results. FEELVOS [33] calculated distance map by local and global matching for better robustness. SiamMask [37] used a depth-wise operation for fast matching and makes a template from a bounding box annotation without accurate annotated mask of a target.

**Consistency Loss:** Consistency loss is widely used for improving performance in semi-supervised learning, enhance robustness from perturbation to input, enable stable training under specific constraints, and so on [10, 21, 46]. In VOS, consistency usually means temporal coherence between neighboring frames by additional clue from optical flow. [32, 35, 41].

## 3 METHOD

In this section, we present our semi-VOS model. Section 3.1 introduces the whole model architecture and how to manage multi-object VOS. Section 3.2 explains the details of template attention module for long-term matching. We also describe how to update the template and how to produce a similarity map. Finally, Section 3.3 demonstrates our temporal consistency loss and how to define new ground truth for mitigating error propagation between neighboring frames.

### 3.1 Overall TTVOS Architecture

We propose a new architecture for VOS as shown in Fig. 2. Our TTVOS consists of feature extraction, template matching, decoding, and template update stages. The template matching is composed of a short-term matching and a long-term matching. The short-term matching enhances localization property by using previous information. This uses a small feature map for producing a coarse segmentation map. However, this incurs two problems: 1) Utilizing only the information of the previous frame causes the output masks overly dependent on previous results. 2) This can not handle shape-changing nor manifest detailed target shape due to a small feature map. To resolve these problems, we propose long-term matching as an adaptive template matching method. This template is initialized from the given first frame condition and updated at each frame. Therefore, it can consider the whole frames and track gradually changing objects. This module uses a larger feature map for getting more detailed information for generating accurate masks. After then, our model executes decoding and updates each templates step by step.

A backbone extracts feature maps $fN_t$ from the current frame, where $fN_t$ denotes a feature map at frame $t$ with an $1/N$-sized width and height compared to the input. Short-term matching uses a small feature map $f16_t$ and the previous frame information for target localization: $f16_{t-1}$ is concatenated with a previous mask heatmap $\hat{H}_{t-1}$, which consists of two channels containing the probability of background and foreground respectively. After then, this concatenated feature map is forwarded by several convolution layers for embedding localization information from the previous frame. This information is blended with $f16_t$ to get an enhanced localization

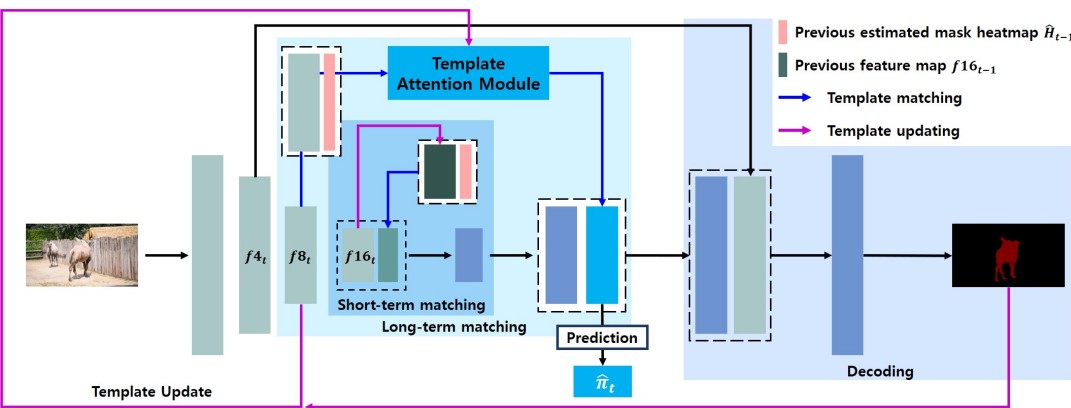

**Figure 2: The overall architecture of TTVOS. A backbone feature is shared in all the processes of TTVOS for efficiency. There are two types of template matching (long-term and short-term), decoding and template update stages in our model. The transition matrix $\hat{\pi}_t$ is computed only in the training phase for enhancing temporal coherence.**

property. In the long-term template matching stage, $f8_t$ is concatenated with the previous mask heatmap, which is compared with the adaptive template to produce a similarity map in the template attention module. The details are in Section 3.2. At only training time, a similarity map estimates a transition matrix to encourage temporal consistency between neighboring frames as detailed in Section 3.3. The resultant similarity map is concatenated with the short-term matching result.

Finally, $f4_t$ is added for a more accurate mask. We use ConvTranspose for upsampling and use PixelShuffle [29] in the final upsampling stage to prevent the grid-effect. After target mask estimation, $f16_t$ and $\hat{H}_t$ are used for updating next short-term template matching, and $f8_t$ and $\hat{H}_t$ are utilized for next long-term template matching. All the backbone features are also shared in the multi-object case, but the stages of two template matching and decoding are conducted separately for each object. Therefore, each object's heatmap always has two channels for the probability of background and foreground. At inference time, all the heatmaps are combined by the soft aggregation method [6, 11].

## 3.2 Template Attention Module

We conjecture that pixels inside a target object have a distinct embedding vector distinguished from non-target object pixels. Our model is designed to find this vector by self-attention while suppressing the irrelevant information of the target object. Each current embedding vector updates a previous long-term template by weighted-average at each frame. After then, the proposed module generates a similarity map by template matching to enhance the detailed region as shown in Fig. 3.

For constructing the current embedding vector, the backbone feature $f8_{t-1}$ and the previous estimated mask heatmap $\hat{H}_{t-1}$ are concatenated to suppress information far from the target object. In Fig. 3, the concatenated feature map is denoted as $X'_{t-1}$. $X'_{t-1}$ is forwarded to two separate branches $f(\cdot)$ and $g(\cdot)$, making $f(X'_{t-1}), g(X'_{t-1}) \in \mathbb{R}^{c_{tp} \times H \times W}$. After then, the feature maps are reshaped to $c_{tp} \times HW$

and producted to generate an embedding matrix $I$ as follows:

$$I = \sigma(f(X'_{t-1}) \times g(X'_{t-1})^T) \in \mathbb{R}^{c_{tp} \times c_{tp}}. \quad (1)$$

Here, $\sigma$ is a softmax function applied row-wise. $I_{i,j}$ is the $(i, j)$ element of $I$, corresponds to an $i$th channel's view about $j$th channel information by dot-producting along $HW$ direction. $X'_{t-1}$ hampers the inflow of information far from the target object by $\hat{H}_{t-1}$. Thus $I_{i,j}$ considers only pixels inside or near the target object, and this operation is similar to global pooling and region-based operation [3] in terms of making one representative value from the whole $HW$-sized channel and concentrating on a certain region. For example, if the hexagon in Fig. 3(a) indicates the estimated location of the target from the previous mask, the information outside of the hexagon is suppressed. Then $f(X'_{t-1})$ and $g(X'_{t-1})$ are compared with each other along the whole $HW$ plane. If the two channels are similar, the resultant value of $I$ will be high (red pixel in Fig. 3(a)); otherwise, it will be low (blue pixel). Finally, we have $c_{tp}$ embedding vectors of size $1 \times c_{tp}$ containing information about the target object. The final long-term template $TP_t$ is updated by weighted-average of the embedding matrix $I$ and the previous template $TP_{t-1}$ as below:

$$TP_t = \frac{t-1}{t}TP_{t-1} + \frac{1}{t}I. \quad (2)$$

The template attention module generates a similarity map $S_t \in \mathbb{R}^{c_{tp} \times H \times W}$ by attending on each channel of the query feature map $q(X_t) \in \mathbb{R}^{c_{tp} \times H \times W}$ through the template $TP_t$ as follows:

$$S_t = TP_t \times q(X_t). \quad (3)$$

In doing so, the previous estimated mask heatmap $\hat{H}_{t-1}$ enhances the backbone feature map $f8_t$ around the previous target object location by forwarding the concatenated feature to a convolution layer resulting in a feature map $X_t$. Then, $X_t$ is forwarded to several convolution layers to generate a query feature map $q(X_t)$ as shown in Fig. 3. In Eq. (3), the similarity is measured between each row of $TP_t$ (template vector) and each spatial feature from $q(X_t)$, both of which are of a length $c_{tp}$. When the template vector is similar to the spatial feature, the resultant $S_t$ value will be high (red pixel in Fig. 3(a)). Otherwise, it will be low (blue in Fig. 3(a)). After then,

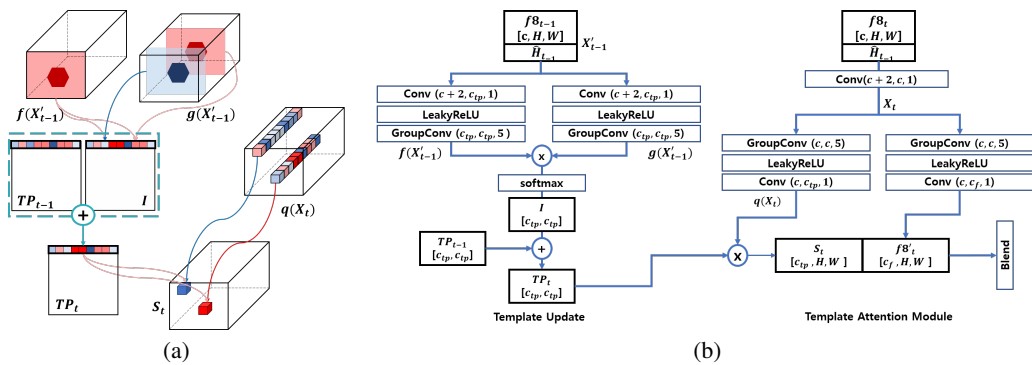

(a)                    (b)

**Figure 3: (a) Process in a template attention module. Here, a red (blue) color means a high (low) similarity between two information. The size of $f(X'_{t-1})$ and $g(X_{t-1})$ is $c_{tp} \times HW$, but we draw feature maps as $c_{tp} \times H \times W$ for the sake of convenient understanding. (b) The detailed structure of a template attention module and a template update. An operation (a,b,c) denotes the input channel, output channel, and kernel size of convolution operation, respectively.**

the global similarity feature $S_t$ and modified feature map $f8'_t$ are concatenated to make the final feature map by blending both results as shown in the bottom of Fig. 3(b).

To reduce computational cost while retaining a large receptive field, we use group convolution (group size of 4) with a large kernel size of $5 \times 5$ for generating $f(\cdot)$, $g(\cdot)$ and $q(\cdot)$. While, depth-wise convolutions cost less than the group convolution, we do not use them because their larger group count adversely impacts the model execution time [19]. We select LeakyReLU as the non-linearity to avoid the dying ReLU problem. We empirically determine that using a point-wise convolution first then applying the group convolution achieves better accuracy (shown in Fig. 3(b)).

Our template attention module has some similarity to GC but is conceptually very different and computationally much cheaper, as shown in Table 1. Unlike GC, which is a memory network approach, our method is a kind of template matching approach. Specifically, GC extracts backbone features again from the new input combining image and mask for generating new memory. Then, it produces a global context matrix by different-sized key and value. However, our template method just combines the current estimated mask and the already calculated backbone feature. Then, we use the same-sized feature maps for self-attention to construct multiple embedding vectors representing various characteristics of the target.

## 3.3 Temporal Consistency Loss

Our adaptive template deals with the target shape-changing problem by analyzing a backbone feature and an estimated mask along the whole executed frames. However, using previous estimation incurs the innate error propagation issue. For example, when the template is updated with a wrong result, this template will gradually lead to incorrect tracking. If the model gets right transition information about how to correct the wrong estimation in the previous frame, the model can mitigate this error propagation problem. For this reason, we calculate a transition matrix $\hat{\pi}_t$ from the output feature map of the template attention module as shown in Fig. 2. We design a novel template consistency loss $L_{tc}$ by $\hat{\pi}_t$, and this loss encourages the model to get correction power and to attain consistency between

|      | Read    | Seg     | Update  | #Param | J&F  |
|------|---------|---------|---------|--------|------|
| GC   | 1.05 G  | 36.8 G  | 37.1 G  | 38 M   | 86.6 |
| Ours | 0.08 G  | 5.29 G  | 0.06 G  | 1.6 M  | 79.5 |

**Table 1: The complexity and accuracy comparison between GC and ours when the input image size is $480 \times 853$. Read, Seg, and updates mean the requirement of FLOPS for reading a memory or a template, making a segmentation mask without a decoding stage, and updating a memory or a template. Our method reduces lots of computations for updating the template.**

neighboring frames:

$$\pi_t = H_t - \hat{H}_{t-1}, \quad L_{tc} = ||\hat{\pi}_t - \pi_t||_2^2. \quad (4)$$

As a new learning target, we make a target transition matrix from ground truth heatmap $H_t$ and previous estimated mask heatmap $\hat{H}_{t-1}$ as in Eq. (4). Note that the first and the second channel of $H_t$ are the probability of background and foreground from a ground truth mask of frame $t$, respectively. By Eq. (4), the range of $\pi_t$ becomes $(-1, 1)$ and $\pi_t$ consists of two channel feature map indicating transition tendency from $t-1$ to $t$. In detail, the first channel contains transition tendency of the background while the second is for the foreground. For example, if the value of $\pi_{t,2}^{i,j}$, the $(i, j)$ element of $\pi_t$ in the second channel, is closer to 1, it helps the estimated class at position $(i, j)$ to change into foreground from frame $t-1$ to $t$. On the other hand, if it is close to $-1$, it prevents the estimated class from turning to the foreground. Finally, when the value is close to 0, it keeps the estimated class of frame $t-1$ for a frame $t$ result.

The reason why we use $\hat{H}_{t-1}$ instead of $H_{t-1}$ is illustrated in Fig. 4. Fig. 4(b) shows ground truth masks, and (c) is the estimated masks at frame $t-1$ (top) and $t$ (bottom). First row of Fig. 4(e) is a visualization of $(H_t - H_{t-1})$ that guides the estimation to maintain the false positive region from the frame $t-1$ to $t$. Second row of Fig. 4(e) is a visualization of $(H_t - \hat{H}_{t-1})$ that guides the estimation to remove false positive region of the frame $t-1$. Fig. 4(d) is marked by blue color for denoting false estimation results comparing between (b) and (c). As shown in Fig. 4(d), the transition matrix $\pi_t$ helps

233
234
235
236
237
238
239
240
241
242
243
244
245
246
247
248
249
250
251
252
253
254
255
256
257
258
259
260
261
262
263
264
265
266
267
268
269
270
271
272
273
274
275
276
277
278
279
280
281
282
283
284
285
286
287
288
289
290

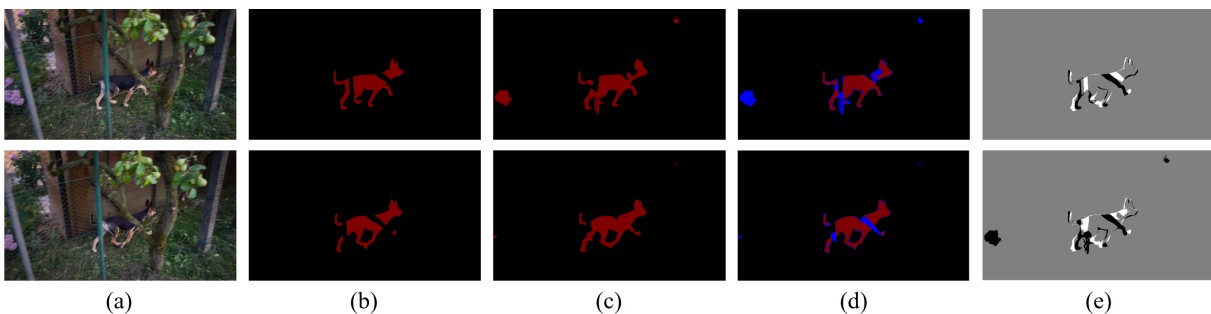

|     |     |     |     |     |
| --- | --- | --- | --- | --- |
| (a) | (b) | (c) | (d) | (e) |

**Figure 4: ((a)-(d)) frame $t-1$ and $t$ from top to bottom. (a) Input image. (b) Ground truth. (c) Our result. (d) Estimated mask with color marking. Blue color means wrong segmentation result, and the blue region in frame $t$ is corrected from frame $t-1$. (e) Visualizing $\pi_{t,2}$. Top: $H_t - H_{t-1}$, Bottom: $H_t - \hat{H}_{t-1}$. $H_t - H_{t-1}$ can not remove false positive region in the top of (c).**

reducing the false positive region from frame $t-1$ to $t$. With $L_{tc}$, the overall loss becomes:

$$Loss = CE(\hat{y}_t, y_t) + \lambda L_{tc}, \qquad (5)$$

where $\lambda$ is a hyper-parameter that controls the balance between the loss terms, and we set $\lambda = 5$. $CE$ denotes the cross entropy between the pixel-wise ground truth $y_t$ at frame $t$ and its predicted value $\hat{y}$.

## 4 EXPERIMENT

Here, we show various evaluations by using DAVIS benchmarks [24, 25]. DAVIS16 is a single object task consisting of 30 training videos and 20 validation videos, while DAVIS17 is a multiple object task with 60 training videos and 30 validation videos. We evaluated our model by using official benchmark code [2]. The DAVIS benchmark reports model accuracy by average of mean Jaccard index $J$ and mean boundary score $F$. $J$ index measures overall accuracy by comparing estimated mask and ground truth mask. $F$ score focuses more contour accuracy by delimiting the spatial extent of the mask.

**Implementation Detail:** We used HRNetV2-W18-Small-v1 [36] for a lightweight backbone network and initialized it from the pre-trained parameters from the official code[3]. We froze every backbone layer except the last block. The size of the smallest feature map is 1/32 of the input image. We upsampled the feature map and concatenated it with the second smallest feature map whose size is 1/16 of the input image. We used ADAM optimizer for training our model. First, we pre-trained with synthetic video clip from image dataset, after then we trained with video dataset with single GPU following [11, 22, 33, 37].

**Pre-train with images:** We followed [16, 22, 40] pre-training method, which applies random affine transformation to a static image for generating synthetic video clip. We used the saliency detection dataset MSRA10K [5], ECSSD [43], and HKU-IS [15] for various static images. Synthetic video clips consisting of three frames with a size of $240 \times 432$ were generated. We trained 100 epochs with an initial learning rate to $1e^{-4}$ and a batch size to 24.

**Main-train with videos:** We initialized the whole network with the best parameters from the previous step and trained the model to video dataset. We used a two-stage training method; for the first

100 epochs, we only used Youtube-VOS with $240 \times 432$ image. We then trained on the DAVIS16 dataset with $480 \times 864$ image for an additional 100 epochs. Both training, we used 8 consecutive frames with a batch size to 8 and set an initial learning rate to $1e^{-4}$.

### 4.1 DAVIS Benchmark Result

**Comparison to state-of-the-art :** We compared our method with other recent models as shown in Table 2. We report backbone models and training datasets for clarification because each model has a different setting. Furthermore, we also show additional results with ResNet50 because some recent models utilized ResNet50 for extracting features.

Our result shows the best accuracy among models with similar speed. Specifically, SiamMask is one of the popular fast template matching methods, and our model has better accuracy and speed than SiamMask on both DAVIS16 and DAVIS17 benchmark. When we used ResNet50, our model has better or competitive results with FRTM-VOS, A-GAME, RANet, and FEELVOS. Also, this ResNet50 based model decreases DAVIS16 accuracy by 2.8% but the speed becomes 1.6 times faster than GC. Therefore, our method achieves favorable performance among fast VOS models and reduces the performance gap from the online-learning and memory network based models.

**Ablation Study :** For proving our proposed methods, we performed an ablative analysis on DAVIS16 and DAVIS17 benchmark as shown in Table 3. SM and LM mean short-term matching and long-term matching, respectively. When we do not use short-term matching or long-term matching, we replaced the original matching method into concatenating the previous mask heatmap and the current feature map. After then the concatenated feature map is forwarded by several convolution layers. Lup represents updating the long-term template at every frame. If not used, the model never updates the template. TC denotes using temporal consistency loss. Without this, the model only uses a cross entropy loss. M denotes using the original ground truth mask for the initial condition; if M is not checked, a box-shaped mask is used for the initial condition like SiamMask. Exp1 is using only short-term matching, and Exp2 is using only long-term matching. Exp3-6 uses both matching methods. Table 3 is the corresponding accuracy for each ablation experiment, and Fig. 6 visualizes efficacy of each template matching.

---

[2]https://github.com/davisvideochallenge/davis2017-evaluation

[3]https://github.com/HRNet/HRNet-Semantic-Segmentation

| Method | Backbone | Model Method | | Train Dataset | | | DV17 | DV16 | FPS |
|---|---|---|---|---|---|---|---|---|---|
| | | OL | Memory | YTB | Seg | Synth | | | |
| OnAVOS [34] | VGG16 | o | - | - | o | - | 67.9 | 85.5 | 0.08 |
| OSVOS-S [20] | VGG16 | o | - | - | o | - | 68.0 | 86.5 | 0.22 |
| FRTM-VOS [26] | ResNet101 | o | o | o | - | - | 76.7 | 83.5 | 21.9 |
| STM [22] | ResNet50 | - | o | o | - | o | 81.8 | 89.3 | 6.25 |
| GC [16] | ResNet50 | - | o | o | - | o | 71.4 | 86.6 | 25.0 |
| OSMN [44] | VGG16 | - | - | - | o | - | 54.8 | 73.5 | 7.69 |
| RANet [40] | ResNet101 | - | - | - | - | o | 65.7 | 85.5 | 30.3 |
| A-GAME [11] | ResNet101 | - | - | o | - | o | 70 | 82.1 | 14.3 |
| FEELVOS [33] | Xception 65 | - | - | o | o | - | 71.5 | 81.7 | 2.22 |
| SiamMask [37] | ResNet50 | - | - | o | o | - | 56.4 | 69.8 | 55.0 |
| **TTVOS (Ours)** | **HRNet** | **-** | **-** | **o** | **-** | **o** | **58.7** | **79.5** | **73.8** |
| **TTVOS-RN (Ours)** | **ResNet50** | **-** | **-** | **o** | **-** | **o** | **67.8** | **83.8** | **39.6** |

**Table 2: Quantitative comparison on DAVIS benchmark validation set. OL and Memory denotes online-learning approach and memory network approach. YTB is using Youtube-VOS for training. Seg is segmentation dataset for pre-training by Pascal [8] or COCO [18]. Synth is using saliency dataset for making synthetic video clip by affine transformation.**

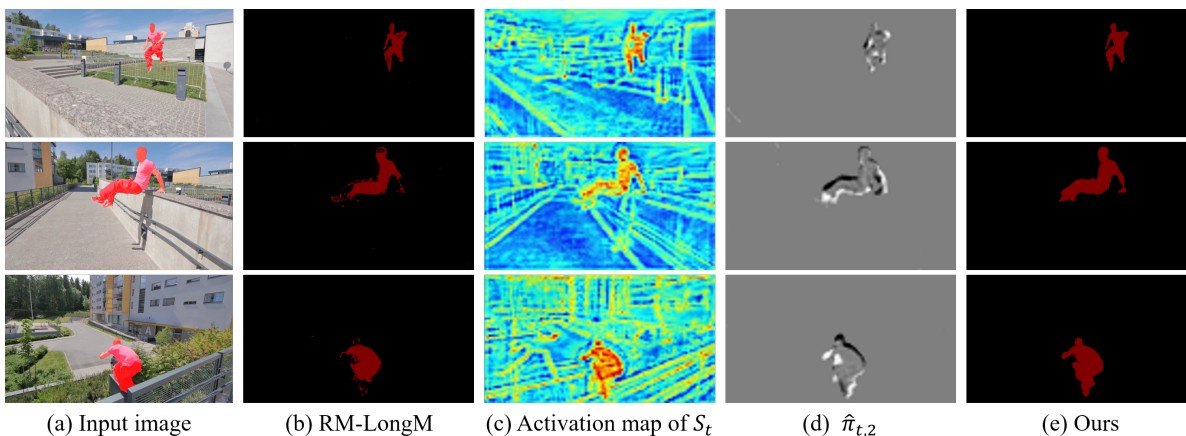

| (a) Input image | (b) RM-LongM | (c) Activation map of $S_t$ | (d) $\hat{\pi}_{t,2}$ | (e) Ours |
|---|---|---|---|---|

**Figure 5: Example of *parkour* for frame 1, 34 and 84 from top to Bottom. Column (a) shows input images overlapped with the ground truth masks. RM-LongM denotes estimated results removing long-term matching information by replacing to zeros.**

| Exp | SM | LM | Lup | TC | M | DV17 | DV16 |
|---|---|---|---|---|---|---|---|
| 1 | o | - | - | - | o | 57.0 | 75.9 |
| 2 | - | o | o | - | o | 54.5 | 78.8 |
| 3 | o | o | o | - | o | 57.5 | 77.1 |
| 4 | o | o | o | o | - | 58.6 | 77.6 |
| 5 | o | o | - | o | o | 57.2 | 77.4 |
| **6** | **o** | **o** | **o** | **o** | **o** | **58.7** | **79.5** |

**Table 3: Ablation study on DAVIS16 and DAVIS17. SM, LM, TC means short-term matching, long-term matching and temporal consistency loss. Lup represents updating long-term template at every frame, and M is using original ground truth mask for initial condition.**

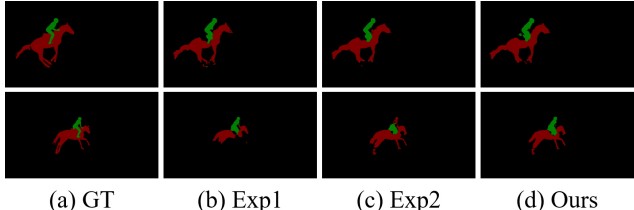

| (a) GT | (b) Exp1 | (c) Exp2 | (d) Ours |
|---|---|---|---|

**Figure 6: *Horsejump-high* example of ablation study for frame 3 and 37 from top to bottom. (a) Ground truth. (b) Using only short-term matching. (c) Using only long-term matching. (d) Our proposed method (Exp6).**

We found that short-term matching helps maintain objects ID from localization clue, and long-term matching improves mask quality by enhancing the detailed regions. For example, Exp1 keeps object ID but fails to make an accurate mask for horse legs, as shown in Fig. 6(b). On the contrary, Exp2 makes accurate shape but loses green-object (rider) ID as shown in Fig. 6(c). Exp2 shows performance degradation on multi-object tracking task (DAVIS 17) due

| | Backbone | DV17 | DV16 |
|---|---|---|---|
| FRTM-VOS [26] | ResNet101 | 76.7 | 83.5 |
| | ResNet18 | 70.2 | 78.5 |
| with TC Loss | ResNet101 | 76.6 | 85.2 |
| | ResNet18 | 71.8 | 82.0 |

**Table 4: DAVIS17 and DAVIS16 results when additional applying temporal consistency loss (TC Loss).**

to failure in maintaining object ID, even it generates more accurate masks than Exp1. Therefore, Exp1 achieves better performance in DAVIS17, and Exp2 shows high accuracy in DAVIS16. Exp3 gets every advantage from both template matching methods, and Fig. 6(d) is our proposed method results (Exp6), which do not lose object ID and generate delicate masks with high performance on both benchmarks.

Exp4-6 explain why our model shows better performance than SiamMask, even using a more lightweight backbone. The initial condition of the box shape mask does not degrade performance a lot comparing with Exp6. However, when the model does not update the long-term template, the accuracy degrades a lot from our proposed method.

**Temporal Consistency Loss :** We conducted further experiments for proving the efficacy of our temporal consistency loss with FRTM-VOS, which is one of the fast online-learning methods, using ResNet101 and ResNet18 for the backbone network. We implemented our proposed loss function based on FRTM-VOS official code[4], and followed their training strategy. Our proposed loss is more useful in the lightweight backbone network (ResNet18) as shown in Table 4. When we applied our loss to the ResNet101 model, the accuracy on DAVIS17 decreased slightly by 0.1%, but it increased 1.7% on DAVIS16. In the ResNet18 model, we improved the accuracy a lot on both DAVIS17 and DAVIS16. We conjecture that using our loss not only improves mask quality but also resolves a problem of overfeating due to fine-tuning by a given condition.

## 5 CONCLUSION

Many semi-VOS methods have improved accuracy, but they are hard to utilize in real-world applications due to tremendous complexity. To resolve this problem, we proposed a novel lightweight semi-VOS model consisting of short-term and long-term matching modules. The short-term matching enhances localization, while long-term matching improves mask quality by an adaptive template. However, using past estimated results incurs an error-propagation problem. To mitigate this problem, we also devised a new temporal consistency loss to correct false estimated regions by the concept of the transition matrix. Our model achieves fast inference time while reducing the performance gap from heavy models. We also showed that the proposed temporal consistency loss can improves accuracy of other models.

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
