# OpenReview forum: "TTVOS: Lightweight Video Object Segmentation with Adaptive Template Attention Module and Temporal Consistency Loss"
_tinyml.org/tinyML/2021/Research_Symposium — tinyML 2021 Regular_

### Official Review · AnonReviewer4 · 2021-01-29

**Overall Merit Score:** 2

**Brief Summary:**

The paper proposes a method for semi-supervised video object segmentation. Specifically, they propose a model based on template matching, which includes short-term and long-term matching. The authors claim that short-term matching is for target object localization, while long-term matching is to improve fine details and handle object shape-changing. A temporal consistency is also proposed to mitigate error-propagation issues. The proposed method achieves reasonable performance on DAVIS16 at a high processing frequency.

**Detailed Comments:**

The results in Table 2 are the central concern for accepting this paper: Besides the unfair comparison for TTVOS with HRNet, TTVOS-RN has a pretty large performance gap with other works with ResNet-50, although it has a higher FPS. However, it is unclear how much of the FPS increase is due to group convolution, and the other methods may achieve similar results in a fair comparison. For a clear acceptance, this paper needs stronger, clearer writing, and a fair comparison of the presented methods.

**Paper Strengths:**


-The authors propose a lightweight VOS model based on template matching by combining short-term and long-term matching.

-The authors design a new consistency loss for semi-VOS without optical flow.

-The presented method achieves reasonable results on DAVIS datasets at a high processing frequency.

-The authors perform a detailed ablation on the individual components of their proposed method and show that the SM + LM + Lup + TC + M combination is necessary for peak performance.


**Paper Weaknesses:**


1.In Table 2, which is the main result of the paper, the TTVOS model is using HRNet as the backbone; while none of previous works are using the same backbone. This is not a fair comparison. The results are notably less substantial for the ResNet50 version (GC has a slightly lower FPS but stronger results).

2.It’s not clear whether the FPS increase is due to the group convolution (Line 254, left). Would the competing methods achieve comparable results with group convolution? I believe an ablation study on group convolutions is necessary to understand the source of the performance increase.

3.The paper could use significant editing for superfluous words, tense consistency, and typos, and grammar. Some examples (there are many throughout the paper):

a.	Phrasing in the abstract and several places throughout the paper is unnecessarily passive, e.g.
i.	This task is tracking => This task tracks
ii.	hard to be utilized => hard to utilize
iii.	template matching methods are devised => we devise template matching methods

b.	Line 19: “temple” => “template”

c.	Figure 4 - I don’t think you need all of those periods and capitalization, you could just have (a) input image (b) ground truth (c) our result

d.	The Method section is a little hard to understand.


**Poster (If Paper Is Rejected):**

1: Yes, ok for poster sesion to nurture work

**Reviewer Confidence:**

4: The reviewer is confident but not absolutely certain that the evaluation is correct

---

### Official Review · AnonReviewer3 · 2021-01-30

**Overall Merit Score:** 3

**Brief Summary:**

This paper proposes an attention-based video segmentation algorithm, which explores the temporal consistency of the video sequence. The experiments show that the proposed method reduced the performance of gaps from heavy models, and the proposed cost function improves the accuracy of the model.

**Detailed Comments:**

- In the loss function (eq.5), two terms are included. One is the pixel-wised cross-entropy between the ground truth and predicted value, the other is the proposed temporal consistency term. However, it’s hard to figure out how you optimize the entire system in the training process, is it end-to-end optimized?

- The performance trade-off measurement (fig.1) could be better illustrated if a curve similar to [20] fig.12 is presented.

**Paper Strengths:**

+ Adding temporal consistency term in the loss function is novel and interesting.

+ With the proposed loss, the accumulation of the error could be avoided.

+ The long term and short term matching could localize the object and refine the details of the object shape.

**Paper Weaknesses:**

- The author should compare the computational complexity with other methods by involving model compression (pruning and quantization) in other methods if possible. Provided that other methods (listed by the author for result comparison) are not targeted for low computational complexity.

- The main weakness of this work lies in the lack of model architecture clarity. Explaining the architecture of the decoder and predictor could help the reader to understand the framework better.

- It would be better to clarify the hardware used for inference time measurements.

**Poster (If Paper Is Rejected):**

1: Yes, ok for poster sesion to nurture work

**Reviewer Confidence:**

4: The reviewer is confident but not absolutely certain that the evaluation is correct

---

### Official Review · AnonReviewer1 · 2021-01-30

**Overall Merit Score:** 3

**Brief Summary:**

This paper presents a lightweight video object segmentation algorithm which uses template matching.  It uses a combination of short-term and long-term modules to reduce complexity.  Results are not state-of-the-art but are competitive relative to the complexity and other performance measures

**Detailed Comments:**

As stated, I'm not familiar with this domain and so my review should be weighted accordingly.  It seems to me that this paper offers a new method of segmentation that gives good results with a significant reduction in model size.  However, as stated previously, the description is entirely disconnected from hardware, and so the constraints of running this algorithm on TinyML-scale hardware are not addressed.  The writing is not great, clearly second language, but the paper is readable and I would not reject it on that score.

**Paper Strengths:**

I'm not familiar enough with this field to judge the novelty, but I would guess that the contribution is moderate.  It does use a clever method for dealing with propagation of error from early predictions.  The reduction in size of the models is significant, one to two orders of magnitude in computation and parameter size.

**Paper Weaknesses:**

The paper seems entirely disconnected from hardware implementation - there is no discussion of accuracy or actual computational methodology.  It is simply a reduction in model size.

**Poster (If Paper Is Rejected):**

1: Yes, ok for poster sesion to nurture work

**Reviewer Confidence:**

2: The reviewer is willing to defend the evaluation, but it is quite likely that the reviewer did not understand central parts of the paper

---

### Decision · Program_Chairs · 2021-02-05

**Decision:**

Accept (Regular)

**Comment:**

Congratulations on your paper's acceptance!

Your paper has been accepted as a full-length regular paper.

Please read the reviews carefully and make sure the concerns are addressed in your final submission.

All accepted papers will be given a slot in the TinyML Summit schedule for an oral presentation on Friday, March 26, 2021.

Camera ready instructions will follow soon. All papers will be hosted on arXiv and published papers will have the following header stamp: “Published as a conference paper at TinyML Research Symposium 2021.” The paper will also be presented on the program website.